# Passive Continuous Variable Measurement-Device-Independent Quantum Key Distribution Predictable with Machine Learning in Oceanic Turbulence

**DOI:** 10.3390/e26030207

**Published:** 2024-02-27

**Authors:** Jianmin Yi, Hao Wu, Ying Guo

**Affiliations:** 1School of Automation, Central South University, Changsha 410083, China; 214612196@csu.edu.cn (J.Y.); 214612208@csu.edu.cn (H.W.); 2School of Computer Science, Beijing University of Posts and Telecommunications, Beijing 100876, China

**Keywords:** continuous variable quantum key distribution, measurement-device-independent, oceanic turbulence model, neural network

## Abstract

Building an underwater quantum network is necessary for various applications such as ocean exploration, environmental monitoring, and national defense. Motivated by characteristics of the oceanic turbulence channel, we suggest a machine learning approach to predicting the channel characteristics of continuous variable (CV) quantum key distribution (QKD) in challenging seawater environments. We consider the passive continuous variable (CV) measurement-device-independent (MDI) QKD in oceanic scenarios, since the passive-state preparation scheme offers simpler linear elements for preparation, resulting in reduced interaction with the practical environment. To provide a practical reference for underwater quantum communications, we suggest a prediction of transmittance for the ocean quantum links with a given neural network as an example of machine learning algorithms. The results have a good consistency with the real data within the allowable error range; this makes the passive CVQKD more promising for commercialization and implementation.

## 1. Introduction

Quantum key distribution is a kind of encrypted means of communication [1,2], which uses the principle of quantum mechanics to enable legitimate parties to exchange secret keys securely. Continuous variable quantum key distribution (CVQKD) has been developed over decades due to its efficient source preparations and compatibility with current devices. Recently, a kind of meliorative protocol called the continuous variable measurement-device-independent (CV-MDI) protocol [3,4] has been proposed, in which a third party Charlie performs Bell state measurement on the quantum states prepared by Alice and Bob, and then broadcasts the result to Alice and Bob to generate the secret key. This detection strategy could counter an attack on practical devices because the measurement is performed by an untrusted third party rather than on Alice or Bob’s side. However, for the classical CVQKD protocol, the quantum states are prepared actively, which requires high precision modulators to reduce modulation error and achieve a complex modulation format, making it expensive for practical implementations.

Currently, a kind of quantum key distribution has been suggested with passive state preparations [5]. Compared with active state preparations, which require high extinction ratio modulators, passive states can be derived from a thermal source for the passive CVQKD. If the initial thermal state generated by the source is strong enough, this scheme can tolerate high detector noise on Alice’s side. Additionally, the output of the source does not need to be single-mode, as an optical homodyne detector can selectively measure a single mode determined by the local oscillator. Since then, passive state preparation has attracted much attention [6,7,8,9]. In 2018, passive states were applied to one-way classical quantum communication [10], and this has been experimentally demonstrated [11,12]. There have been many results of passive state preparations in recent years, such as security analysis [13] and applications [14]. In 2019, passive states were used for the CV-MDI QKD protocol [15].

Over time, the CV-MDI system has expanded from the free space channel to the ocean quantum links [16,17,18]. However, in the implementation of the ocean quantum links, many factors, such as seawater salinity, oceanic turbulence, and chlorophyll concentration, have an affect on the propagation of light beams [19]. To solve these difficulties, we propose a machine learning-based prediction of ocean transmittance to provide data reference for engineering applications in practice. In recent years, in the field of QKD, machine learning has been paid more and more attention. In 2020, Z. A. Ren et al. employed machine learning methods to select an optimal QKD protocol [20]; in the same year, a random forests algorithm was used to directly predict the optimal parameters of the QKD system [21]. Two years later, Zhou et al. used neural networks to construct a secure key rate prediction model for discrete modulation continuous variable systems [22]. In 2023, Ahmadian. M et al. used machine learning to improve the polarization tracking compensation scheme of a QKD system [23]. The organization of this paper is as follows. In Section 2, CV-MDI QKD with passive state preparation is suggested. In Section 3, we analyze the characteristics of the oceanic channel and propose a machine learning-assisted model based on an oceanic turbulence model for transmittance prediction. In Section 4, the secret key rate in the oceanic scenario is derived. Section 5 shows the simulation results, and then Section 6 draws the conclusions.

## 2. CV-MDI QKD with Passive State Preparation

The Gaussian-modulated coherent states (GMCS) QKD protocol is implemented based on the prepare-and-measure scheme. And from Eve and Bob’s points of view, the state from Alice is a single-mode thermal state with an average photon number of a half of modulation variance. In fact, the security of the GMCS QKD is commonly proved based on an equivalent entanglement-based protocol [24], where Alice performs conjugate homodyne detection on one mode of a two-mode squeezed vacuum state and sends the other mode to Bob. In this picture, the state from Alice is indeed thermal.

Here, we prepare the passive state by using a thermal source. There is a relationship between the value of the number of photons output at the Alice terminal and the modulation variance V in the GMCS QKD protocol, and the protocol with passive states requires a Gaussian modulator with a modulation variance of V. The preparation of passive states is implemented by taking advantage of a thermal source, beam splitters, optical attenuators, and homodyne detectors rather than the amplitude and phase modulators. The CV-MDI QKD protocol with passive state preparation is depicted in Figure 1, and its implementation can be described as follows.

Step 1: Alice and Bob each prepare a thermal sources. They use a 50:50 beam splitter to split the optical signal into two correlated spatial modes (the average number of photons output by each source is $n_{0}$), denoted by (${\mathrm{Mod}}_{A1}$, ${\mathrm{Mod}}_{A2}$) (for Alice’s side) and (${\mathrm{Mod}}_{B1}$, ${\mathrm{Mod}}_{B2}$) (for Bob’s side), respectively. Next, Alice (Bob) attenuates the average photon number of ${\mathrm{Mod}}_{A1}$ (${\mathrm{Mod}}_{B1}$) down to a half of the variance of $V_{A}$ ($V_{B}$) by using an optical attenuator. The modulated signals are then transmitted to a third party, Charlie.

Step 2: Alice (Bob) performs heterodyne detection on both the X and P quadratures of mode ${\mathrm{Mod}}_{A2}$ (${\mathrm{Mod}}_{B2}$). They broadcast the measurement results to Charlie. The quadratures of ${\mathrm{Mod}}_{A1}$ (${\mathrm{Mod}}_{B1}$) at Charlie’s side have the relation with ${\mathrm{Mod}}_{A2}$ (${\mathrm{Mod}}_{B2}$) as follows ($X_{A1}=\sqrt{\frac{2\eta_{A}}{\eta_{D}}}X_{A2}$, $P_{A1}=\sqrt{\frac{2\eta_{A}}{\eta_{D}}}P_{A2}$) and ($X_{B1}=\sqrt{\frac{2\eta_{B}}{\eta_{D}}}X_{B2}$, $P_{B1}=\sqrt{\frac{2\eta_{B}}{\eta_{D}}}P_{B2}$). Here, $\eta_{A}$ and $\eta_{B}$ represent the transmittance of the attenuator, while $\eta_{D}$ represents the efficiency of the practical homodyne detector.

Step 3: Charlie mixes the received ${\mathrm{Mod}}_{A1}$ and ${\mathrm{Mod}}_{B1}$ on a balanced beam splitter and conducts Bell state measurement on them. The results are detected by conjugate homodyne detection at the output ports. After that, Charlie broadcasts the quadratures ($X_{C}$,$P_{C}$) over a classical public channel to Alice and Bob.

Step 4: After repeating these steps several times, Alice and Bob obtain a string of raw keys. Next, they apply post-processing operations such as privacy amplification and error correction to filter the data of ($X_{A1}$, $P_{A1}$), ($X_{B1}$, $P_{B1}$), and ($X_{C}$, $P_{C}$). Then, Alice and Bob get the final secret keys. The process is similar to the traditional CV-MDI QKD protocol with active state preparation, where communication parties can obtain final secret keys if the detected total noise falls below a certain threshold value. Compared with the Gaussian state, the passive state does not require the participation of a high-precision Gaussian modulator, which reduces the complexity and cost of the system.

## 3. Transmittance Prediction with Machine Learning

In this section, we first analyze the effect of the oceanic turbulence channel on light propagation, then we suggest a machine learning-based prediction model that can be used as a reference for practical underwater quantum communication systems.

### 3.1. Optical Propagation Characteristics of the Oceanic Turbulence Channel

Based on the seawater chlorophyll model and the elliptical model, which have been described in [25]—with the exception of the seawater extinction coefficient *T*—the Monte Carlo method used in the elliptic beam model for the oceanic turbulence channel has general applicability to any other ocean. The seawater extinction coefficient is the sum of the ocean absorption factor $t_{\mathrm{abs}}$ and scattering factor $t_{\mathrm{sca}}$, which have an effect on the absorption and scattering of light in the ocean. $t_{\mathrm{abs}}$ and $t_{\mathrm{sca}}$ are functions of the ocean depth *d* and wavelength λ, the specific function varies depending on the type of ocean, and we have analyzed the optical propagation characteristics in ocean type S1. Mathematically, $t_{\mathrm{abs}}$ and $t_{\mathrm{sca}}$ has the form:(1)$$\begin{array}{cc} \; & t_{\mathrm{abs}}=l_{c}^{0}{\left[u_{c}\left(d\right)\right]}^{0.602}+l_{w}+l_{f}^{0}u_{f}\left(d\right)e^{-k_{f}\lambda }+l_{h}^{0}u_{h}\left(d\right)e^{-k_{h}\lambda },\\ \; & t_{\mathrm{sca}}=m_{s}^{0}u_{s}\left(d\right)+m_{l}^{0}u_{l}\left(d\right)+m_{w}, \end{array}$$
where $l_{w}$ represents the absorption due to pure water in relation to wavelength λ, and $l_{w}$, corresponding to different wavelengths, is given by [26]. $l_{h}^{0}$ denotes the absorption coefficient of chlorophyll α in relation to λ, and $l_{h}^{0}$ corresponding to different wavelengths is given by [27]. The details of Equation (Equation 1) are given in Appendix A. By fitting the function, we get the functional relationship between $l_{w}$, $l_{h}^{0}$, and λ, respectively. It should be noted that this function does not contain quantum noise; parameters like the absorption coefficient of chlorophyll α, the loss of light propagation in pure water, etc. are all related physical factors that affect light propagation, and they quantified the effect of seawater on the propagation of light.

The relationship between wavelengths, depth, absorption factor, and scattering factor are given in Figure 2a,b.

The characteristics of S1 can be clearly seen in Figure 2. The absorption and scattering factors of the S1 ocean increase significantly at 100–150 m due to high chlorophyll concentration and plankton enrichment at this depth, which leads to a sharp decrease in the secret key rate near this depth, as detailed in Section 5.

### 3.2. Transmittance Prediction of Seawater Channel

In practice, the estimation of transmittance and excess noise requires the two legitimate parties to sacrifice part of the raw keys for the parameter estimation procedure; the more the raw keys are consumed, the more accurate the estimation of the transmittance and the excess noise is. However, sacrificing too many raw keys will affect the efficiency of communication. Meanwhile, the estimation for the transmittance in the parameter estimation is intended to estimate its lower bound as much as possible to ensure the absolute security of communication. In this scheme, we do not discard the parameter estimation step. Instead of the transmittance obtained from the parameter estimation step, we use the transmittance predicted by machine learning to participate in the estimation of the secret key rate; the former is more accurately close to the true value than the lower bound of the transmittance ($T_{low}$) and thus we can obtain a higher secret key rate without sacrificing more raw keys.

This approach allows the CVQKD system to maintain stable performance in various environmental conditions, thereby improving the system’s reliability. The structure of the Elman neural network is illustrated in Figure 3, and the prediction procedure is outlined in Figure 4.

The Elman neural network is a type of recurrent neural network (RNN) that is used for time series prediction and sequence modeling [28]. It has a feedback loop that allows information from previous time steps to be fed back into the network, enabling it to capture temporal dependencies in the input sequence.

We present below an overview of its structure and training process. An Elman network typically consists of three main layers: an input layer, which receives external inputs and feeds them into the next layer; a hidden layer, which is a set of neurons that perform computations based on both current input and a context vector received from the previous time step. Each neuron has self-recurrent connections along with feedforward connections from the input layer; an output layer, which processes the outputs of the hidden layer neurons to generate the final output; context units—a distinctive characteristic of the Elman network is the context layer or ’memory’ units, which are a copy of the hidden layer activations at one time step, which are then fed back into the hidden layer during the next time step. This feedback loop enables the network to maintain some form of short-term memory that can influence its future predictions.

There are basic steps for training an Elman network. Initialization: Assign random initial weights and biases to all connections between layers; Forward Propagation: For each sequence step, input the current time step data into the input layer. The hidden layer computes its activations based on the current input and the previously stored context. The output layer generates its prediction using the hidden layer activations; Backpropagation through time (BPTT): After making predictions for an entire sequence, calculate the loss function comparing predicted outputs to target values across the sequence. BPTT extends standard backpropagation by unrolling the network over time and computing gradients through the unfolded network; Calculate the error gradient for each time step and update the weight matrices and bias vectors accordingly; Parameter Update: Using an optimization algorithm like stochastic gradient descent (SGD) or variants such as Adam, update the network parameters according to the computed gradients, aiming to minimize the total sequence loss; Iterative Training: Repeat this process over many iterations (epochs) until the performance on a validation set stabilizes or starts to degrade, indicating convergence or potential overfitting; Regularization: If necessary, apply regularization techniques to control model complexity and prevent overfitting. During the training process, it is crucial to monitor the learning curves, adjusting hyperparameters such as learning rate, batch size, and the number of hidden units if needed, to ensure efficient and accurate learning of temporal patterns in the data.

The GA-Elman algorithm [29] is a hybrid approach that combines the Elman recurrent neural network with an adaptive genetic algorithm to optimize the network’s parameters for time series prediction. The main idea is to use the genetic algorithm to search for the optimal combination of weights and biases in the Elman network to minimize the prediction error. The algorithm starts by initializing the Elman network with random weights and biases. The training data are then fed into the network, and the output is computed. The genetic algorithm is used to optimize the weights and biases based on the prediction error. The genetic algorithm creates a population of candidate solutions, which are evaluated based on their fitness, i.e., how well they minimize the prediction error. The fittest solutions are selected for reproduction, and their offspring inherit their genetic traits through crossover and mutation. The Elman network is trained using the optimized weights and biases, and the process is repeated until the prediction error converges or the maximum number of iterations is reached. The trained network is then used to predict future values of the time series. The GA-Elman algorithm has several advantages over other time series prediction methods. It can handle nonlinear and non-stationary time series, and it can adapt to changing environments. The genetic algorithm allows for a global search of the parameter space, which can lead to better solutions than gradient-based methods.

The relationship between $t_{abs}$, $t_{sca}$ and transmittance *T* is $T=e^{-(t_{abs}+t_{sca})z}$ [30], *z* is the transmission distance. Combined with Equation (Equation 1), we can find that the transmittance is a binary function whose independent variables are depth and transmission distance, so the inputs to the machine learning model are the depth and transmission distance.

The transmittance prediction of the Elman and GA-Elman algorithms on transmission can be seen in Figure 5. To provide a quantitative analysis of the performance improvements made to the Elman algorithm, we present the prediction errors of GA-Elman and Elman for 1200 sets of test samples, as shown in Figure 6. Prediction error refers to the difference between the value of the transmittance output of the predicted model and the real value when we input the depth and distance.

The numerical analysis demonstrates that this model is capable of predicting the fluctuation of transmittance within an acceptable error range, which has a strong correlation with the actual transmission values. The prediction results can be used to assist actual derivation and calculation under certain circumstances.

The elliptic model provides the probability density function (PDF) of the transmittance, and an estimate of the transmittance is obtained by solving the inverse function of the cumulative distribution function, but the value of the actual measured transmittance can be any arbitrary value within the range of the PDF, and is not exactly equivalent to the former. According to [25], the variance of the transmittance is estimated to be on the order of $10^{-5}$, with a transmittance of at least 0.4 or higher at effective underwater communication distances. Therefore, when the model predicts the transmittance based on actual measurements, the error in the model’s prediction is within acceptable limits, even if the actual value of the transmittance at the predicted location is the value of the transmittance corresponding to a very small probability in the PDF.

The protocol performance under the transmittance prediction model based on machine learning is shown in Figure 7.

The black dashed line represents the secret key rate curve after the transmittance predicted by machine learning is applied to the parameter estimation, and the green dashed line represents the secret key rate curve when the parameter estimation takes the lower bound of the transmittance without the application of the machine learning model.

## 4. Security Analysis

According to the above-mentioned processing, we obtain the transmittance in seawater channels, and hence we can establish the correlation among average transmittance, ocean depth, and transmission distance. Moreover, the passive state for the CV-MDI QKD protocol usually leads to excess noise, which provides an opportunity for eavesdropping through joint attacks. From [3], we assume that Eve adopts the most general joint attack against the protocol, which involves using the joint two-mode attack strategy that targets both links simultaneously. This approach is considered more effective than a single-mode attack strategy, which involves an additional layer of complexity to the security analysis.

### 4.1. Secret Key Rate in Asymptotic Scenarios

To begin with, we assume that the preparation sides have the same variance, thus the covariance matrix (CM) can be written as $V_{A_{1}B_{1}|C_{1}}=V_{A_{1}⊕B_{1}}-ZC^{-1}Z^{T}$, where $V_{A_{1}⊕B_{1}}$ represents the reduced covariance matrix (CM) of Alice and Bob’s modes, while *C* denotes the outcome CM of Charlie, and *Z* represents the complex correlations between these CMs. Regarding the eavesdropping strategy, assuming a Gaussian distribution, Eve has the potential to intercept the traveling modes $A_{1}$ and $B_{1}$, which are mixed with two quantum-correlated ancillary modes. The reduced state $V_{E_{1}E_{2}}$ can be written in the normal form:(2)$$V_{E_{1}E_{2}}=\left(\begin{array}{cc} \varpi_{1}I_{2} & G\\ G & \varpi_{2}I_{2} \end{array}\right);$$
$\varpi_{1}$ and $\varpi_{2}$ are the variance of the thermal excess noise disturbing the corresponding link, with $I_{2}=\mathrm{diag}\left(1,1\right)$, with G=diag(g,−g), where
(3)$$g=\min\left[\sqrt{\left(\varpi_{1}-1\right)\left(\varpi_{2}+1\right)},\sqrt{\left(\varpi_{1}+1\right)\left(\varpi_{2}-1\right)}\right]$$
is set to minimize the secret key rate. From [3,31], the simplified covariance matrix between Alice and Bob can be calculated as:(4)$$V_{A_{1}B_{1}|C}=\left[\begin{array}{cc} \left(V-\frac{T_{A}\left(V^{2}-1\right)}{\vartheta }\right)I_{2} & \frac{\sqrt{T_{A}T_{B}}\left(V^{2}-1\right)}{\vartheta }\sigma_{Z}\\ \frac{\sqrt{T_{A}T_{B}}\left(V^{2}-1\right)}{\vartheta }\sigma_{Z} & \left(V-\frac{T_{B}\left(V^{2}-1\right)}{\vartheta }\right)I_{2} \end{array}\right],$$
with $\sigma_{Z}=\mathrm{diag}\left(1,-1\right)$, $V=V_{A}+1=V_{B}+1$, and
(5)$$\vartheta =V\left(T_{A}+T_{B}\right)+\varpi_{1}\left(1-T_{A}\right)+\varpi_{2}\left(1-T_{B}\right)-2g\sqrt{\left(1-T_{A}\right)\left(1-T_{B}\right)}.$$
Let $X_{E}$ denote the information that Eve can get by using the two-mode attack, and it is given by
(6)$$X_{E}=S\left(\rho_{A_{1}B_{1}|C}\right)-S\left(\rho_{B_{1}|C\alpha }\right),$$
where $S\left(\rho_{A_{1}B_{1}|C}\right)$ and $S\left(\rho_{B_{1}|C\alpha }\right)$ can be calculated as:(7)$$\begin{array}{cc} \; & S\left(\rho_{A_{1}B_{1}|C}\right)=H\left(\lambda_{1}\right)+H\left(\lambda_{2}\right),\\ \; & S\left(\rho_{B_{1}|C\alpha }\right)=H\left[\sqrt{\det\left(V_{A_{1}B_{1}|C}\right)}\right], \end{array}$$
with $H\left(x\right)=\frac{1+x}{2}\log_{2}\left(\frac{1+x}{2}\right)-\frac{x-1}{2}\log_{2}\left(\frac{x-1}{2}\right)$. Here, $\lambda_{1}$ and $\lambda_{2}$ are the symplectic eigenvalues of $V_{A_{1}B_{1}|C}$. The mutual information between Alice and Bob is given by:(8)$$I_{AB}=\log_{2}\frac{V}{X_{total}},$$
where $X_{total}$ can be divided into $X_{total}=X_{loss}+\varepsilon_{E}$. The pure loss in channel from senders to Charlie is defined as $X_{loss}$, which has the form $X_{loss}=2\frac{T_{A}+T_{B}}{T_{A}T_{B}}$, and the total excess noise $\varepsilon_{E}=\varepsilon_{P}+\varepsilon_{0}$, where $\varepsilon_{0}$ is the background noise, and $\varepsilon_{P}$ is the total excess noise in the process of passive state preparation. Therefore, we have:(9)$$\begin{array}{c} \begin{array}{c} \varepsilon_{P}=\varepsilon_{A}+\varepsilon_{B},\\ \varepsilon_{A}=\frac{2\;V_{A}}{\eta_{D}n_{0}}\left(1+V_{el}\right)-\frac{V_{A}}{n_{0}},\\ \varepsilon_{B}=\frac{2\;V_{B}}{\eta_{D}n_{0}}\left(1+V_{el}\right)-\frac{V_{B}}{n_{0}}, \end{array} \end{array}$$
where $V_{A}$ and $V_{B}$ are the modulation variance, $V_{el}$ is the electronic noise of the homodyne detector, $\eta_{D}$ is the efficiency of the homodyne detector, and $n_{0}$ is the average number of photons output by the thermal source.

### 4.2. Secret Key Rate in the Finite-Size Case

In the finite-size condition, the secret key rate is given by [32,33]:(10)$$K=\frac{n}{N}\left[K_{\infty }\left(T_{A}^{low},T_{B}^{low},\varepsilon_{X_{C}}^{high},\varepsilon_{P_{C}}^{high}\right)-\Delta \left(n\right)\right],$$
where the signals exchanged by Alice and Bob are *N*. Due to the effects of finite size, Alice and Bob should conduct the parameter estimation by using a number of *m* keys in the practical condition. The remaining number *n*, which has the correlation with *m*, is given by n=N−m, which is used to generate the secret key. The correction term Δ(n) is simplified as:(11)$$\Delta \left(n\right)=7\sqrt{\frac{\log_{2}2/\epsilon_{P_{A}}}{n}}.$$

The estimation of error in privacy amplification $\epsilon_{P_{A}}$ is set to $10^{-10}$. The noise terms of $X_{C}$ and $P_{C}$ has the form:(12)$$\begin{array}{cc} \varepsilon X_{C}=\varepsilon P_{C}= & \varepsilon_{P}+\frac{1}{2}\left[\varpi_{1}\left(1-T_{A}\right)+\varpi_{2}\left(1-T_{B}\right)\right]-g\sqrt{\left(1-T_{A}\right)\left(1-T_{B}\right)}, \end{array}$$
(13)$$\delta_{\varepsilon X_{C}}=\delta_{\varepsilon P_{C}}=\sqrt{\frac{2\varepsilon_{X_{C}}}{m}}.$$
The maximum noise of $X_{C}$ and $P_{C}$ generated in Charlie’s detection is given by:(14)$$\begin{array}{cc} \; & \varepsilon_{X_{C}}^{high}=\varepsilon_{X_{C}}+6.5\delta_{\varepsilon X_{C}},\\ & \varepsilon_{P_{C}}^{high}=\varepsilon_{P_{C}}+6.5\delta_{\varepsilon P_{C}}. \end{array}$$
Considering the security of the protocol, the channel transmittance is considered in the worst case because Alice and Bob are the same as Charlie and, consequently, only the case between Alice and Charlie is introduced. The expression can be conducted from:(15)$$T_{A}^{low}=\frac{1}{2}\left(X_{2}^{low}-X_{1}^{up}\right),$$
with
(16)$$\begin{array}{cc} \; & X_{2}^{low}=X_{2}-6.5\sqrt{\mathrm{Var}\left(X_{2}\right)},X_{1}^{up}=X_{1}+6.5\sqrt{\mathrm{Var}\left(X_{1}\right)},\\ \; & X_{1}=\langle T_{A}\rangle -{\langle \sqrt{T_{A}}\rangle }^{2},X_{2}=\langle T_{A}\rangle +{\langle \sqrt{T_{A}}\rangle }^{2},\\ \; & \mathrm{Var}\left(X_{1}\right)=\mathrm{Var}\left(X_{2}\right)=\sigma_{\langle T_{A}\rangle }^{2}+2\sigma_{\langle \sqrt{T_{A}}\rangle }^{4}\left[1+2\frac{\mu^{2}\sqrt{T_{A}}}{\sigma_{\langle \sqrt{T_{A}}\rangle }^{2}}\right]. \end{array}$$
Here, we have notations $\sigma_{\langle T_{A}\rangle }^{2}=\int {\left(P\left(T_{A}\right)\right)}^{2}\sigma_{{\hat{T}}_{A}}^{2}$, $\sigma_{\langle \sqrt{T_{A}}\rangle }^{2}=\int {\left(P\left(T_{A}\right)\right)}^{2}\mathrm{Var}\left(\sqrt{{\hat{T}}_{A}}\right)$, and $\mu_{\sqrt{T_{A}}}=\int P\left(T_{A}\right)E\left(\sqrt{{\hat{T}}_{A}}\right)$. Taking $\sqrt{{\hat{T}}_{A}}=\sqrt{\frac{2{\left({\hat{C}}_{AC}\right)}^{2}}{\eta V_{A}^{2}}}$; we obtain
(17)$$\begin{array}{cc} \; & \delta_{\hat{T_{A}}}^{2}=\frac{\mathrm{Var}\left({\hat{C}}_{AC}\right)8T_{A}}{\eta V_{A}^{2}},\\ \; & E\left(\sqrt{{\hat{T}}_{A}}\right)=\sqrt{T_{A}},\\ \; & \mathrm{Var}\left(\sqrt{{\hat{T}}_{A}}\right)=\frac{2\mathrm{Var}\left({\hat{C}}_{AC}\right)}{\eta V_{A}^{2}}. \end{array}$$
The variance of ${\hat{C}}_{AC}$ is $\frac{\left(\eta V_{A}^{2}T_{A}+V_{A}V_{N}\right)}{m}$ and $V_{N}$ is the variance of $X_{N}$.

## 5. Simulation Results

In what follows, we demonstrate the performance of the CV-MDI QKD system in terms of the secret key rate and transmission distance. In numerical simulations, we set the average number of output photons to 800 per pulse and the modulation variance to 60. For simplicity, we assume that the homodyne detector is noiseless and has an efficiency of 0.95 [15]. The communication block size is $10^{8}$, and thermal noise $\varpi_{1}$∼$\varpi_{2}$∼ 1.01. Depending on the distance between Alice and Bob, and Charlie, it can be classified as asymmetric and symmetric. It is worth mentioning that the system performs better in the asymmetric case when Alice and Charlie are closer to each other than in the symmetric case. Therefore, we will only present the performance in the former case.

The variation relationship between the secret key rate as the dependent variable and the transmission distance and depth as the independent variables is shown in Figure 8.

The title ’distance’ in Figure 8 means the distance between Alice and Bob, namely the effective distance between two underwater communication parties. The green dashed line represents the change of the secret key rate with depth when the communication distance between the two communication parties is 1.65 m, so that the change of the dent of the three-dimensional surface in a certain depth segment can be more clearly seen, which just validates the previous analysis of the extinction coefficient, that is, the extinction coefficient increases sharply in this depth segment, and naturally the corresponding secret key rate should decline sharply in this depth segment.

Additionally, we find how the secret key rate correlates with depth and transmission distance. The green curves demonstrate that the impact of ocean depth on the secret key rate is significant for a given transmission distance. This phenomenon is attributed to the presence of a strong optical fading effect at a specific depth in the ocean, which is denoted in Section 3. Therefore, it is advisable to avoid deploying communication devices in areas where the extinction factor is concentrated. The red curve illustrates the accepted phenomenon that the secret key rate decreases with increasing transmission distance for a given depth. The trend of the secret key rates in the finite-size case is similar to that in ideal conditions. However, the rates are lower due to the effects of the finite size.

## 6. Conclusions

We have proposed passive state CV-MDI to ocean scenarios. Then, we analyzed the optical propagation characteristics of the oceanic turbulence channel; moreover, we have presented a transmittance prediction model using the GA-Elman neural network. This model exhibits a high level of predictive accuracy for quantum communication in oceanic turbulence. The machine learning-assisted CVMDI protocol with passive states has improved its performance, although, limited by the complexity of the underwater environment and the attenuation of light propagation, the transmission distance has not been significantly improved, which is the limitation of this paper. However, the method for improving the lower bound of transmittance in parameter estimation by predicting transmittance is also suitable for free-space channels. In the future, with the proposal of quantum communication protocols with higher performance, it is expected to provide a new idea for auxiliary QKD systems.

The secret key rates in the asymptotic and finite-size cases are derived and the performance of the scheme is calculated. These above findings contribute to the advancement of passive CV-MDI QKD in challenging underwater environments. The ability to accurately predict transmittance in oceanic turbulence can enhance the security and reliability of quantum communication systems operating in such conditions.

## Figures and Tables

**Figure 1 entropy-26-00207-f001:**
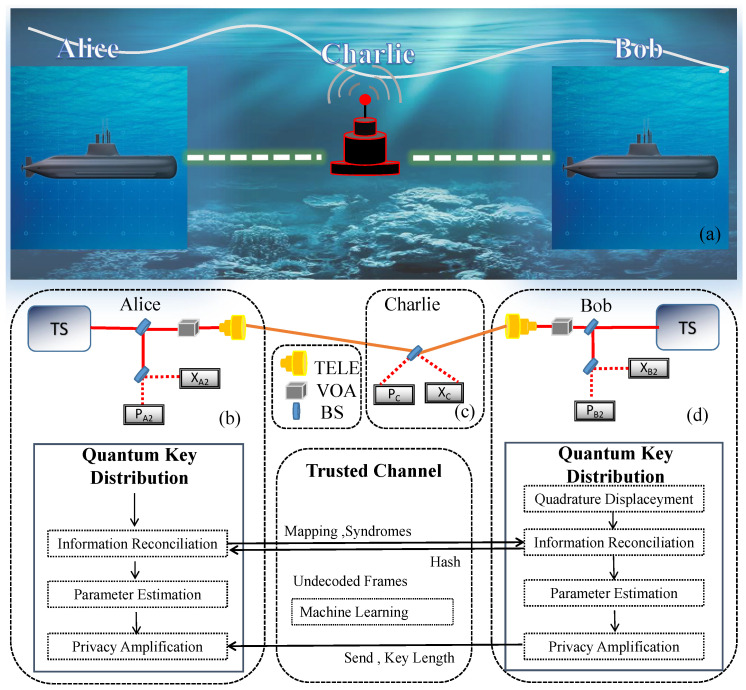
(**a**) Schematic diagram of the application of the underwater CV-MDI protocol. (**b**) Alice’s side. (**c**) Charlie’s side. (**d**) Bob’s side. In the data processing stage, the machine learning module is used to predict transmittance. The specific explanation of this algorithm can be found in part 3. VOA, variable optical attenuator; TS, thermal source; the red dashed line denotes conjugate homodyne detection. TELE, telescope.

**Figure 2 entropy-26-00207-f002:**
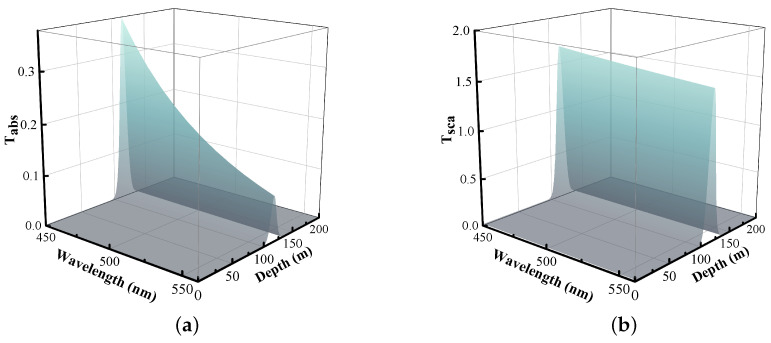
Variation of absorption and scattering factors with wavelength and depth. (**a**) Absorption factor. (**b**) Scattering factor.

**Figure 3 entropy-26-00207-f003:**
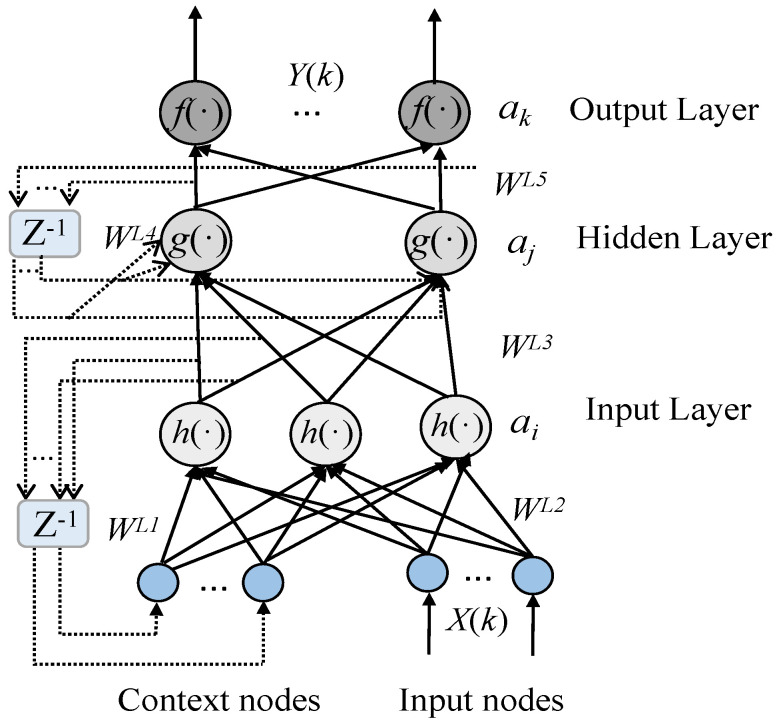
Structure of the Elman neural network. X(k), Y(k), input and output vectors; $W^{Li}$ (where i = 1, 2,...…, 5), connection weights; h(·),g(·),f(·), the nodal activation functions; $a_{i},a_{j},a_{k}$, the thresholds; $Z^{-1}$, the unity delay.

**Figure 4 entropy-26-00207-f004:**
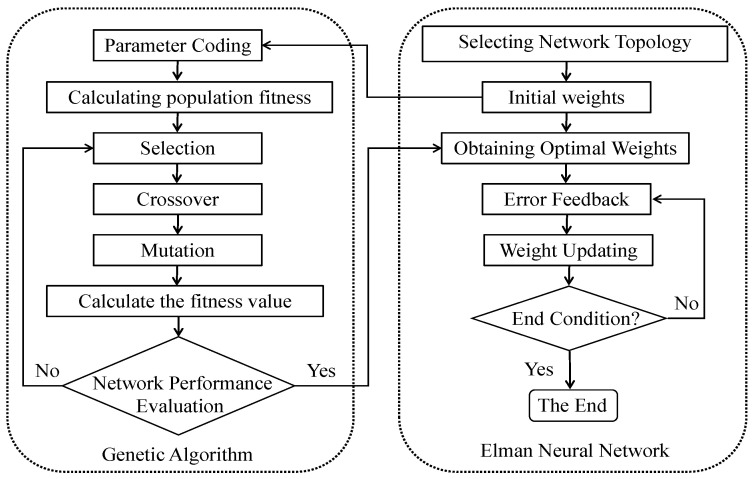
Flowchart of GA-Elman algorithm.

**Figure 5 entropy-26-00207-f005:**
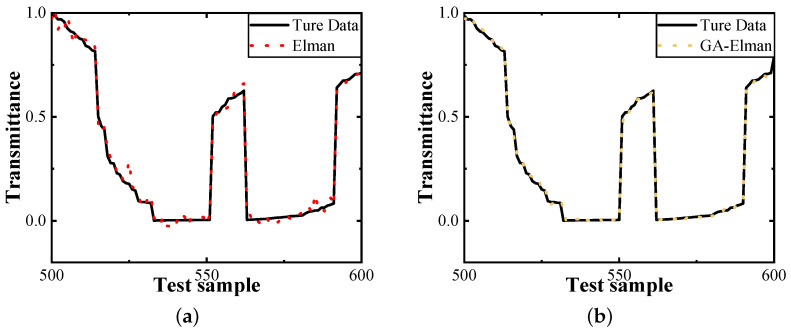
Predictions for transmittance. (**a**) Performance of Elman algorithm. (**b**) Performance of GA-Elman algorithm.

**Figure 6 entropy-26-00207-f006:**
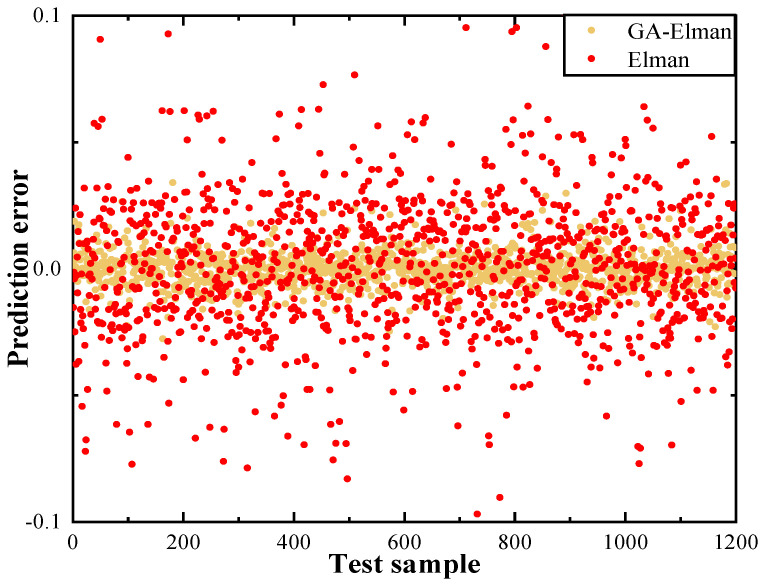
Prediction error of Elman and GA-Elman algorithm. The average absolute percentage error between Elman and GA-Elman are 2.814% and 0.506%, respectively.

**Figure 7 entropy-26-00207-f007:**
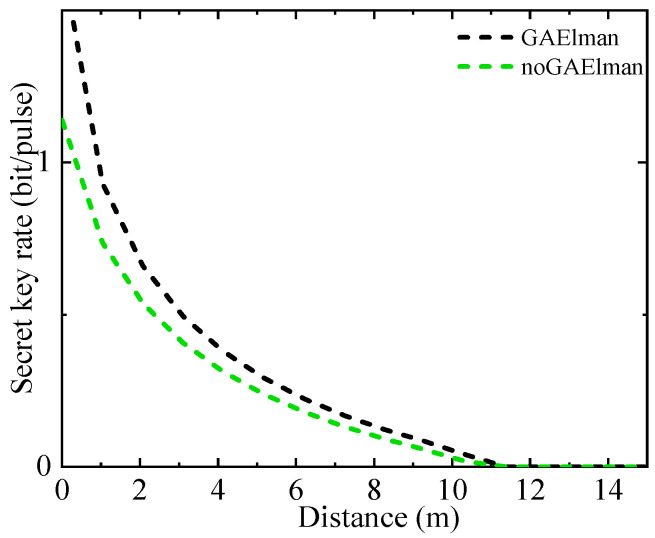
Performance improvement diagram of QKD system assisted by machine learning model.

**Figure 8 entropy-26-00207-f008:**
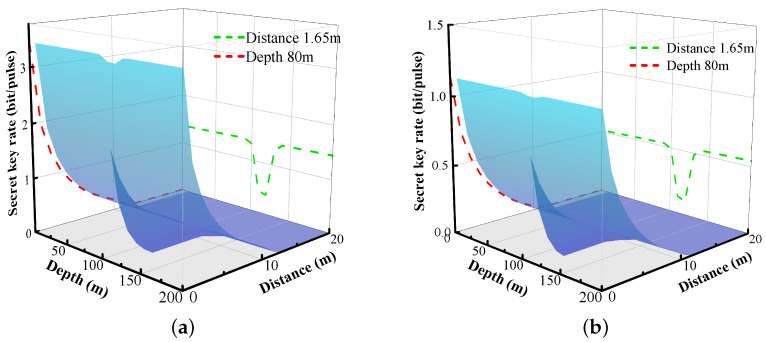
Secret key rate in asymmetric case.(**a**) Asymptotic case. (**b**) Finite-size case.

## Data Availability

Data are contained within the article.

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
