# Peer review of "Passive Continuous Variable Measurement-Device-Independent Quantum Key Distribution Predictable with Machine Learning in Oceanic Turbulence"

_entropy, 2024, doi:10.3390/e26030207_

Round 1

Reviewer 1 Report

Comments and Suggestions for Authors

The authors  describe the protocol that containing the machine learning part;  Introduce the machine learning method and how it implement into the protocol; Showing the results, including the key rate, security and the simulation result;

The protocol:

Step 1: Alice and Bob generate source with photon number 2,use beam splitter to get (,) for Alice and (,) for Bob. Then modulate () by an optical attenuator with variance () then send () to Charlie.

Step 2: Alice and Bob perform measurement on () using hetero-dyne detection method. Then send the measurement result ()to Charlie.

Step 3: Charlie conducts Bell measurement on () then board-cast the output (,) to Alice and Bob. (,) is the elements of raw key. Repeat Step1-3 several times, Alice and Bob could gain the raw key.

Step 4: Post-processing operation, assuming that the quantum channel noise is under a threshold. The operation including error correction and private operation such as privacy amplification using the data  (,) , (,) and (,) and error correction. The author say then the final key is generated.

Which step will the machine learning take part in is not introduced. So the suggestion is to have few more steps to describe in detail.

And there are some points that seems not clear or not complete.

1. In section 3, the paper firstly introduced a function of oceanic turbulence coefficients that influencing optical propagation undersea. My first question is would this function containing quantum noise?

2. Also in section 3, the paper then introduced machine learning method. By showing structure of Elman neural network and GA-Elman algorithm and say it is useful with the evidence given by test sample. I have to say it is not clear or not complete. What is the input data X and output data Y? Why the protocol needs machine learning? The advantages? In what reason to use neural network?

3. In section 5, the title 'distance' in Figure 7 is confusing. If it means the distance between Charlie and Alice(Bob), the data is unacceptable for the described protocol. Because this paper assume the Alice and Bob to be the underwater vehicle, the distance 1.65m or 0-20m is so low that Alice and Bob seems unnecessary to use in QKD practical.

4. Some characters in this paper are not well introduced such as X, P.

This paper need carefully fix, there could be more than the questions mentioned before. And please be careful on the physical reality when considering the practical environment and practical protocol.

Author Response

Dear editor, thank you for your careful review. The review comments are uploaded in the attachment, please see the attachment, thank you.

Reviewer 2 Report

Comments and Suggestions for Authors

 - As general comments the authors should state in what do this work improve the state of the art. Specifically, what does machine learning offers over current methods currently used to estimate the channel parameters in QKD. They should also reference some other works of Machine Learning applied to QKD to contextualize their work.

- Regarding this affirmation: "By utilizing measured transmittance data to predict the variations in transmittance, it is possible to eliminate the need for parameter estimation, thereby improving the secret key rate."  The authors should clarify how can they guarantee mathematically that their neuronal network substitutes parameter estimation.

- The authors state: "In numerical simulations, we set the average number of output photons to 800 and the modulation variance to 60." They should clarify what they refer to, is it photons per symbol? Per second? In this case they have a classical signal, not quantum.

- At any point in the manuscript do the authors mention whether the they use gaussian or discrete modulation. They should clarify on this.

- In section 4 they mention “excesive noise”, where they should really say “excess noise” as it is the correct term.

- They also mention that joint attacks are the optimum ones but they do not include any reference to back this argument.

- In equation 9 there is a long mathematical development using the function erfinv but no citation to this is found either.

- They use a noiseless detector and a detection efficiency of 0.95 for their simulations. These are not realistic values so they should comment on why these are used, in case more realistic values do not achieve any positive SKR as we can assume.

- They should also specify more details about the neuronal network, how they train it, and specially what the “prediction error” they mention is.

- They should also extend their conclusions. The results are very limited in distance, they should explain why this is and how to improve it in the future.

Comments on the Quality of English Language

The overall level of the English is good.

Author Response

(The authors gave the same response as above.)

Reviewer 3 Report

Comments and Suggestions for Authors

This theoretical work is devoted to the development of a new approach based on machine learning technique to predict channel characteristics of the continuous variable key distribution in the water quantum network. Authors consider a passive state preparation scheme which offers several simplifying benefits for underwater implementation. Authors predict that their model for quantum links based on Genetic Algorithm-Elman neural network approach will be of practical usefulness for the prediction of quantum signal transmittance in underwater quantum communications. In my view, authors present a novel approach to solve for the optimal combination of parameters (wights and biases) to minimize errors. This method can handle nonlinear and non-stationary time series, and it can adapt to changing environments like ocean depth and transmission distance.

The paper is clearly written, with figures effectively supporting the presented results. I recommend the manuscript for publication in Entropy in the present form.

Comments on the Quality of English Language

English requires minor revision.

Author Response

(The authors gave the same response as above.)

Round 2

Reviewer 2 Report

Comments and Suggestions for Authors

All comments were properly addressed.